# Engineered amphiphilic peptides enable delivery of proteins and CRISPR-associated nucleases to airway epithelia

Sateesh Krishnamurthy[1], Christine Wohlford-Lenane[1], Suhas Kandimalla[1], Gilles Sartre[1], David K. Meyerholz [2], Vanessa Théberge[3], Stéphanie Hallée[3], Anne-Marie Duperré[3], Thomas Del'Guidice [3], Jean-Pascal Lepetit-Stoffaes[3], Xavier Barbeau[3], David Guay[3] & Paul B. McCray Jr. [1]*

The delivery of biologic cargoes to airway epithelial cells is challenging due to the formidable barriers imposed by its specialized and differentiated cells. Among cargoes, recombinant proteins offer therapeutic promise but the lack of effective delivery methods limits their development. Here, we achieve protein and SpCas9 or AsCas12a ribonucleoprotein (RNP) delivery to cultured human well-differentiated airway epithelial cells and mouse lungs with engineered amphiphilic peptides. These shuttle peptides, non-covalently combined with GFP protein or CRISPR-associated nuclease (Cas) RNP, allow rapid entry into cultured human ciliated and non-ciliated epithelial cells and mouse airway epithelia. Instillation of shuttle peptides combined with SpCas9 or AsCas12a RNP achieves editing of *loxP* sites in airway epithelia of ROSA$^{mT/mG}$ mice. We observe no evidence of short-term toxicity with a widespread distribution restricted to the respiratory tract. This peptide-based technology advances potential therapeutic avenues for protein and Cas RNP delivery to refractory airway epithelial cells.

[1] Department of Pediatrics, Pappajohn Biomedical Institute, University of Iowa, Iowa City, IA 52242, USA. [2] Department of Pathology, University of Iowa, Iowa City, IA 52242, USA. [3] Feldan Therapeutics, Québec, QC, Canada. *email: paul-mccray@uiowa.edu

The airway epithelium is a critical interface between the host and environment and a common site of genetic and acquired disease states such as asthma, chronic obstructive pulmonary disease, cystic fibrosis (CF), and infectious diseases. Modifying airway epithelia with therapeutic proteins, gene expression cassettes, or genome-editing reagents offers great promise, but efficacious delivery is a common challenge. This is because the respiratory epithelium presents barriers through its specialized cell types, secreted host defense factors, and muco-ciliary transport[1]. While in vivo delivery with vector systems has advanced[2–7], airway epithelia remain poorly transduced by many viral and non-viral approaches[8–12].

The direct transfer of recombinant proteins is a promising alternative to the use of nucleic acids to deliver antibodies[13] growth factors, anti-viral peptides[14], or CRISPR-associated nuclease (Cas) ribonucleoprotein (RNP)[15]. RNPs comprise Cas nucleases and guide RNAs (gRNA), which are of interest for therapeutic gene editing because of their rapid effect and transient nuclease activity[16]. While appealing, protein delivery is hampered by the limited availability of safe and effective delivery methods[17,18]. Cas RNPs contain negatively charged gRNA that enables their delivery using cationic lipids or electroporation in responsive cell types[19]. However, other proteins, such as green fluorescent protein (GFP), antibodies, or luciferase, are poorly delivered using such procedures[20]. The lack of appropriate bio-degradable lipids[21] and the difficulty in applying electroporation in vivo[22,23] limit options for non-viral Cas RNP delivery to air-ways in vivo.

Cell-penetrating peptides (CPPs) are short peptides generally carrying a net positive charge that independently mediate inter-nalization of molecules across cell membranes (e.g., HIV TAT (human immunodeficiency virus *trans*-acting activator of tran-scription) peptide[24] and penetratin[24,25]). However, CPP-mediated delivery often results in limited cytosolic distribution due to endosomal sequestration of cargoes[26–29]. To avoid endo-somal entrapment, CPP may be combined with endosomolytic peptides such as endosomal leakage domains, which bind and transiently destabilize endosomal membranes[30–34].

We recently developed a CPP-ELD system for in vitro delivery of proteins and Cas RNPs to multiple cell types[35]. Based on the study of CPP-ELD peptide sequences[32,33,36], we designed addi-tional peptide-based agents, here termed shuttles, that further improve delivery of Cas RNP to hard-to-transduce human nat-ural killer (NK) cells[35,37]. We demonstrate here that next-generation shuttles enable protein transfer to another refractory cell type by delivering reporter proteins and Cas RNPs in vitro and in vivo to differentiated epithelial cells lining the airways despite the physicochemical barriers imposed by these cells.

## Results

### Shuttle peptides enable delivery of Cas RNP to NK cells.
The first-generation shuttles were designed by fusing the endosomo-lytic peptide CM18 and the CPP PTD4 with six-histidine tags, at one or both ends of the peptide (CM18-PTD4)[35]. The activity of CM18-PTD4 constructs was demonstrated by delivering different protein cargoes to multiple cell types. We then further modified peptide sequences to increase their delivery efficacy[37]. Here, we present a subset of three rationally designed peptides termed Shuttle10 (S10), Shuttle18 (S18), and Shuttle85 (S85). Figure 1a shows peptide sequences aligned with CM18-PTD4. While the peptides share features, including an improved hydrophobic cluster derived from the CM18 domain and an optimized hydrophilic/cationic tail derived from the cell-penetrating PTD4 peptide, they diverge in others. In addition to their variable length (25–34 amino acids), S10 contains five highly hydrophobic

residues in the CM18-derived domain, while S18 and S85 have eight, the S85 linker was reduced to a single glycine, and the PTD4-derived domain of S18 contains six cationic residues compared to four in S10 and S85. Using these three rationally designed peptides, we demonstrate improved editing of NK cells when co-incubated with Cas12a RNP for 90 s. Figure 1b presents the editing measured at the *NKG2A* locus following Cas12a RNP delivery to NK cells. RNP delivery by S10, S18, or S85 improved editing, achieving indels of 25%, 23%, and 26%, respectively, compared to the previously reported CM18-PTD4 that enabled 10% editing[35].

### Shuttle peptides deliver protein to airway epithelial cells.
As shuttle peptides enabled Cas12a RNP delivery to NK cells, we hypothesized that they could deliver proteins and Cas RNPs to hard-to-transduce well-differentiated primary cultures of human airway epithelial (HAE) cells (Fig. 1c). As differences in peptide length, hydrophobicity, linker, and charge may influence delivery efficiency when applied to new cell types, we first test the feasi-bility of peptide-mediated delivery in HAE with CM18-PTD4, S10, S18, and S85 using GFP protein. Shuttle peptides were combined with recombinant nuclear-targeted GFP protein and applied to the apical surface for 15 min, and then the GFP signal was quantified by fluorescence imaging (Fig. 1d, left panel). At a 20 μM concentration, S10 achieved the greatest GFP delivery. En face images demonstrated GFP signal throughout the epithelium (Fig. 1d, right panel). Similar results were obtained when the same four shuttle peptides were used to transfer GFP protein to primary cultures of porcine tracheal epithelia, a large animal model of CF[38,39] (Supplementary Fig. 1). To identify the cell types transduced in human airway epithelia, we co-localized GFP and specific cell type markers. Fifteen minutes after applying S10 and GFP, we observed GFP-positive (GFP+) nuclei in cells positive for acetylated α-tubulin, a ciliated cell marker (Fig. 1e, left panel, white arrows). GFP also co-localized with non-ciliated cells (Fig. 1e, left panel, white arrowheads) and Muc5AC, a marker of goblet (secretory) cells (Fig. 1e, right panel, white arrowhead). By morphometric analysis, the S10 transduction efficiency ranged from 27 to 35% for all cell types of the surface epithelium (32 ± 2.0, mean ± SE, n = 3). We also applied the S10 peptide and GFP to the apical surface of freshly excised human tracheal tissue explants. One hour later, GFP+ nuclei co-localized with ciliated and non-ciliated surface epithelia (Fig. 1f).

### Shuttle peptides deliver Cas RNPs to airway epithelia.
As a proof of principle, we delivered Cas12a RNP targeting *CFTR* intron 22–23 to HAE from non-CF donors with the four shuttle peptides used to deliver GFP. This intronic region is the site of a *CFTR* splicing mutation termed 3849 + 10C>T that introduces a premature termination codon and causes CF[40] (see diagram in Fig. 2a). We assessed Cas12a RNP-induced indels using the Surveyor assay and quantified by Sanger sequencing 3 days after delivery (Fig. 2b). We observed an indel frequency of 9–26%, with S10 conferring the most efficient Cas12a RNP delivery. Figure 2c, d shows the effects of S10 dose and duration of incubation on editing efficiency. While increasing the peptide concentration improved editing, the duration of incubation did not. To inves-tigate the editing efficiency of Cas12a RNPs for another target, we selected the *HPRT1* locus (Fig. 2e). S10 and S85 achieved the greatest indel% (Fig. 2e). We also tested a Cas9 RNPs designed to *CFTR* exon 11 in non-CF epithelia (Fig. 2f). *CFTR* exon 11 is the site of the common F508del mutation. The CM18-PTD4, S18, S10, and S85 peptides achieved similar indel%. To illustrate the difficulty in delivering macromolecular cargo to HAE, we trans-fected Cas9 and Cas12a RNPs with three commercial Lipofection

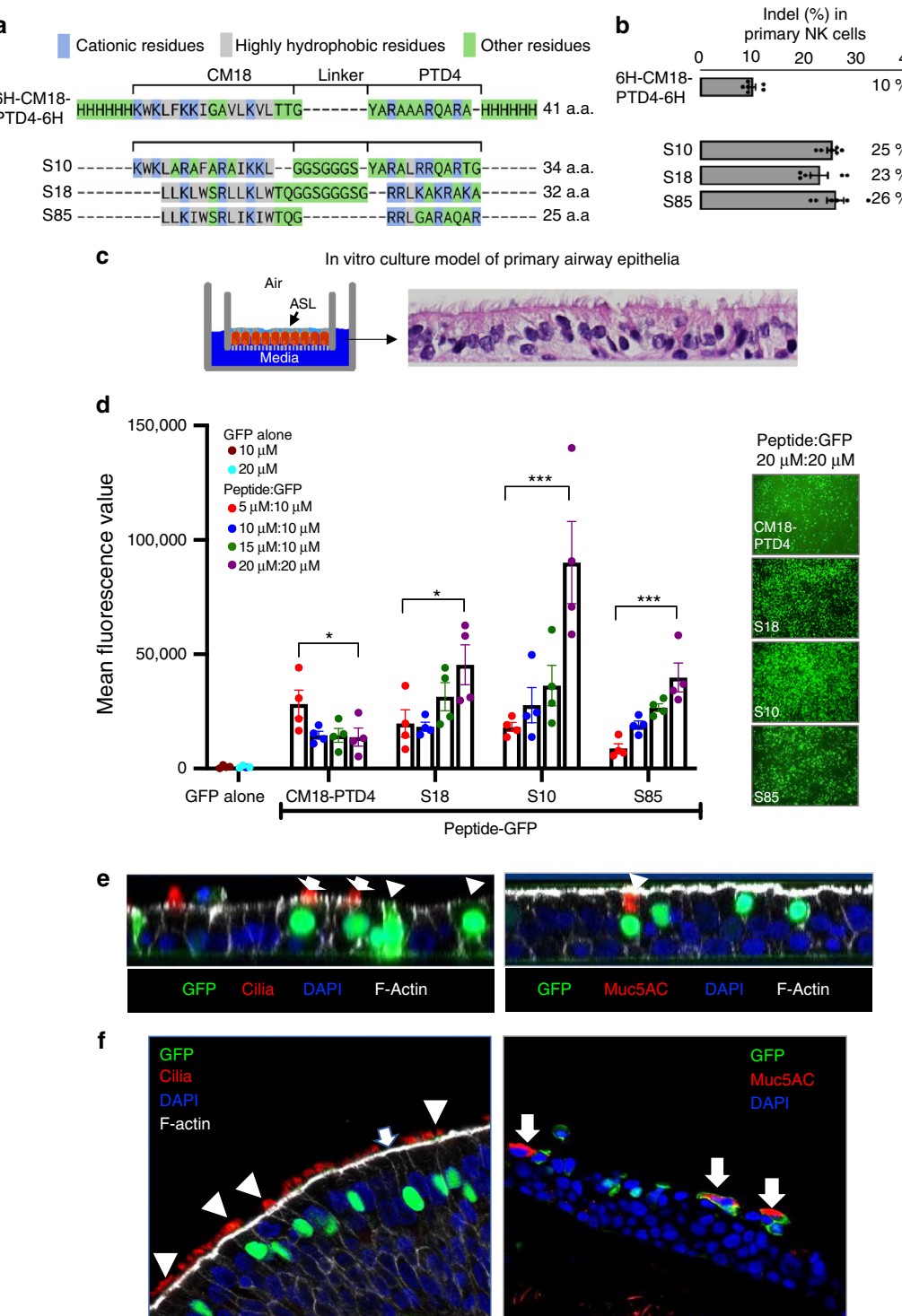

reagents and observed no editing of two different *CFTR* loci (Supplementary Fig. 2).

**Shuttle peptides deliver protein in vivo.** To investigate the utility of shuttle peptides in vivo, we delivered nuclear-targeted GFP protein to the airways of adult BALB/c mice by nasal instillation. We selected the S10 peptide based on its performance in HAE (Fig. 1). We administered S10 and GFP once or twice over an 8 h period. Approximately 18 h after the final treatment, mice were euthanized and GFP distribution evaluated. We observed widespread nuclear GFP signal within large

and small airway epithelia (Fig. 3a) and co-localization confirmed that GFP was delivered to ciliated and non-ciliated secretory (club) cells (Fig. 3b, c). These represent the two predominant surface cell types of the murine large (Fig. 3b) and small (Fig. 3c) airways. Non-ciliated club cells are a well-characterized progenitor cell type[41]. In the distal lung, we observed occasional GFP⁺ cells that co-localized with SP-C, a marker of alveolar type II cells (Fig. 3d, white arrows). The transduction efficiency in the large airways was 27 ± 8% and 46 ± 5% (mean ± SE) for one or two doses, respectively. In the small airways, the delivery efficiency was 32 ± 1% and 30 ± 2% (mean ± SE) with one or two doses, respectively (Fig. 3e).

**Fig. 1** Shuttle peptide design and protein delivery to airway epithelia. **a** Amino acid sequences of shuttle peptides. Sequences aligned to highlight structural similarities. Cationic residues are highlighted in blue; hydrophobic residues are in gray. Remaining residues are in green. **b** Indel% in primary NK cells following Cas12a RNP delivery targeting *NKG2A* gene with indicated shuttle peptide ([Cas12a]: 1.33 μM; [crRNA]: 2.0 μM). Results quantified 48 h after delivery (mean ± SE; n = 3 independent experiments). **c** Shuttle peptide-mediated delivery of nuclear-targeted GFP protein to human airway epithelia (HAE) and tracheal explants. Left panel: Schematic of primary HAE grown at air–liquid interface. ASL: airway surface liquid. Right panel: photomicrograph of well-differentiated primary HAE culture demonstrating pseudostratified columnar epithelium. Hematoxylin and eosin stain, ×40 magnification. **d** Efficiency of GFP delivery by different shuttle peptides. Peptides applied at four concentrations (5, 10, 15, or 20 μM); GFP concentrations 10 or 20 μM. Left graph: Cells were transduced with indicated peptide and GFP combinations and GFP quantified by fluorescence reader. Results mean ± SE; n = 4 donors. *P < 0.05, **P < 0.005, ***P < 0.001, by one-way ANOVA with Tukey's multiple comparison test. Right panels: Representative en face views of epithelial sheets following GFP delivery at the highest peptide:GFP concentration (20 μM:20 μM), ×20 magnification. Formulations were applied for 15 min. **e** GFP localization in multiple epithelial cell types following shuttle peptide-mediated delivery. HAEs were transduced for 15 min with GFP protein and S10 peptide. Left panel: confocal image (XZ) shows co-localization of GFP and specific marker of ciliated cells (α-tubulin, red, indicated by white arrows), non-ciliated cells (white arrowheads), F-actin (phalloidin stain, gray), nuclei (DAPI, blue). Right panel: co-localization of GFP and goblet cell marker (Muc5AC, red, indicated by white arrowhead), nuclei (DAPI, blue), and F-actin (phalloidin stain, gray). n ≥ 6 different donor epithelia; ×64 magnification. **f** Shuttle peptide delivery of GFP to human tracheal explants. S10 (20 μM) and GFP protein (20 μM) applied to tracheal explant for 1 h. Left panel: both ciliated (α-tubulin, red, white arrowhead) and non-ciliated (white arrow) cells were transduced. Right panel: non-ciliated goblet cells were transduced (Muc5AC, red, white arrows); ×40 magnification. Data underlying this Figure are provided as Source Data file

**Shuttle peptides enable in vivo gene editing with Cas9 RNP.** To test the feasibility of RNP-mediated gene editing in vivo, we used ROSA$^{mT/mG}$ mice in which all cells express membrane-localized tdTomato fluorescence from the Rosa26 locus[42]. Deletion of the tdTomato cassette by Cre-lox recombination or by dual Cas RNP cleavage results in the expression of cell membrane-localized GFP fluorescence[42] (Fig. 4a). Due to the absence of a Cas12a protospacer adjacent motif (PAM) in the *loxP* region and previous positive outcomes using Cas9 in Cre-lox reporter mice[43], we next evaluated Cas9 RNPs in vivo.

We delivered Cas9 RNPs targeting both *loxP* sites into the airways of ROSA$^{mT/mG}$ mice using the S10 peptide. Mice received one peptide-Cas9 RNP dose per day on two consecutive days (see Fig. 4a). One week following the second dose, we localized GFP-expressing edited cells in lung tissues. Similar to GFP protein delivery, we observed widespread RNP-mediated editing in epithelial cells of the large (Fig. 4b, c) and small airways (Fig. 4d). Both ciliated and non-ciliated cells expressed GFP (Fig. 4e, arrows, arrowheads), confirming editing of the two predominant surface cell types. In the distal lung, we observed occasional co-localization of GFP and surfactant protein C (SP-C), a marker of alveolar type II cells (Fig. 4f, arrows). Following delivery of Cas9 RNP alone there was no editing in large or small airways (Fig. 4g, h). The editing efficiency in the large airways was 13 ± 2% (mean ± SE). In the small airways, the editing efficiency was 12 ± 1% (mean ± SE) (Fig. 4i).

We next asked whether shuttle peptide-mediated protein delivery was associated with acute toxicity. Application of S10 alone or with GFP protein had no effect HAE viability as assessed by lactate dehydrogenase (LDH) release (Supplementary Fig. 3). To address in vivo toxicity, we delivered two doses of Cas9 RNP and the S10 peptide, S10 alone, or phosphate-buffered saline (PBS) to mice as described above. At 1 and 7 days post delivery, we quantified inflammatory cell infiltrates by bronchoalveolar lavage (BAL) total and differential cell counts, lung tissue cytokine and chemokine messenger RNA (mRNA) abundance, and histopathological evaluation. We observed no significant differences in BAL cell counts and neutrophil or macrophage abundance between PBS-treated animals and those receiving S10 alone or S10 and Cas9 RNP (Fig. 5a). Histopathologic scoring demonstrated no evidence of increased pulmonary inflammation or injury in any group (Fig. 5b). Transient increases in chemokine (C-C motif) ligand 5, interleukin-15, and interleukin-10 mRNA were observed in the peptide-Cas9 RNP treatment group 1 day after delivery (Supplementary Fig. 4). These findings resolved by day 7.

To investigate cargo fate after lung deposition, we screened shuttle peptides for delivery of firefly luciferase protein to HAE (Supplementary Fig. 5). We selected the CM18-PTD4 peptide as it was most effective in vitro. One hour after luciferase protein delivery to airways of BALB/c mice with the CM18-PTD4 peptide, we measured luciferase distribution using bioluminescent imaging. Bioluminescence localized to nasal and pulmonary tissues, consistent with protein retention in the respiratory tract (Supplementary Fig. 5).

**Shuttle peptides facilitate Cas12a delivery in vivo.** The optimal Cas12a PAM sequence, TTTV[44], is absent from the *loxP* regions of ROSA$^{mT/mG}$ mice. Nevertheless, we tested Cas12a RNP delivery targeting a CTTC motif in the *loxP* sequence (see Fig. 6a). Success using this suboptimal CTTC PAM was previously demonstrated[45]. Using S10, we delivered Cas12a RNPs and achieved editing in large (Fig. 6b, c) and small airway (Fig. 6d) epithelia with a distribution similar to that observed with Cas9 RNPs. Both ciliated and non-ciliated cell types were edited (Fig. 6e). In the distal lung, we observed occasional co-localization of GFP and SP-C, indicating editing of alveolar type II cells (Fig. 6f). The editing efficiency in the large airways was 12 ± 1% (mean ± SE); in the small airways the editing efficiency was 10 ± 1% (mean ± SE) (Fig. 6g).

## Discussion

There is great interest in protein and nucleic acid-based therapeutics for airway diseases, but delivery to respiratory epithelial cells remains exceedingly challenging. Several viral vectors[3,6,46], non-viral liposomes[21,47,48], and other nanoparticle formulations are in development to address delivery challenges[49,50], but none have advanced to widespread clinical utility. While some viral vectors overcome delivery barriers[4,5,51], they are complex biologics and genetic payloads may exceed the capacity of some vectors. Non-viral DNA delivery approaches were also used in airway epithelia but remain inefficient, may elicit inflammatory responses following aerosol delivery[47,52–54], or lead to endosomal or lysosomal entrapment[8,11]. The use of Cas RNP rather than coding mRNA or DNA offers advantages in a therapeutic context by limiting genome exposure to editing machinery and decreasing off-target events[16]. With relatively good agreement, these studies suggest that engineered peptides confer effective and non-toxic protein and Cas9 or Cas12a RNP transfer into airway epithelia in vitro and in vivo.

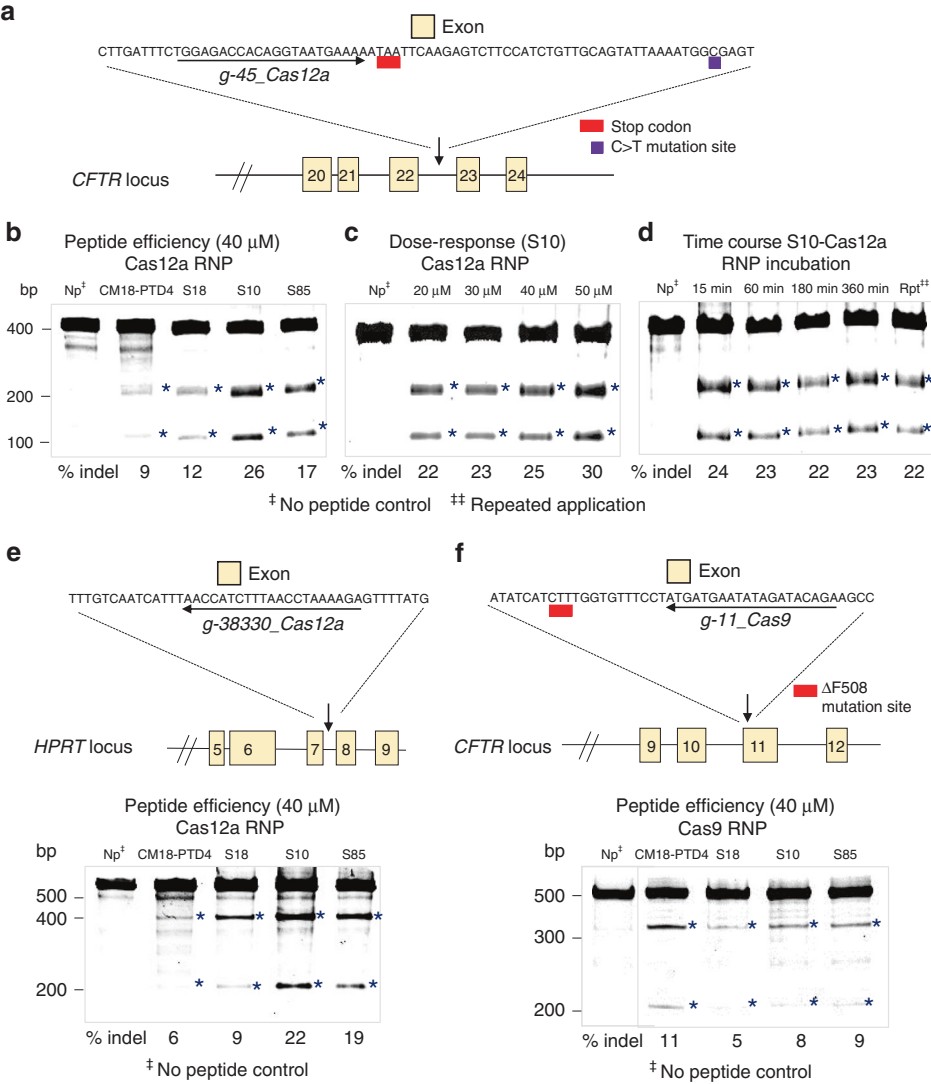

**Fig. 2** Shuttle peptides deliver Cas12a and Cas9 RNPs to HAE. **a** Schematic showing *CFTR* locus in region of 3849 + 10C>T mutation (not to scale) and the sequence of the Cas12a guide RNA target. **b** Editing at the *CFTR* locus following delivery of Cas12a RNPs using four different peptides. Shuttle peptides were tested for Cas12a RNP delivery using gRNA targeting *CFTR* intron 22–23. Materials were applied for 15 min, cells were harvested 72 hr later for Surveyor assay; Indel% determined by Sanger sequencing. Asterisks denote bands observed with gene editing. Np indicates Cas12a RNP with no peptide. **c** S10 peptide dose–response on Cas12a RNP editing of *CFTR* locus. HAE transduced with fixed RNP concentration [Cas12a]: 1.33 μM; [gRNA]: 2 μM and S10 peptide concentrations varied (20–50 μM). Cells incubated with peptide-RNP for 15 min, and harvested 72 h later for Surveyor assay (Control: Cas12a RNP alone). **d** Effect of incubation time and repeated of peptide-Cas12a RNP delivery on editing. [S10]: 40 μM; [RNP]: 40 μM, applied for indicated times. After 72 h, cells prepped for Surveyor assay and Sanger sequencing (Np indicates Cas12a RNP with no peptide, incubated for 3 h; Rpt denotes repeated application of peptide/RNP × 3 daily doses). $n = 3$ donors. **e** Schematic of *HPRT1* locus and Cas12a guide RNA target sequence along with editing efficiency on delivery of RNPs. Screen of four peptide formulations at 40 μM concentration, [RNP]: 2.5 μM; [gRNA]: 2.0 μM on primary HAE. Indicated peptide-RNP applied for 3 h; 72 hr later, cells were processed for Surveyor assay. Asterisks denotes genome editing. $n = 3$ donors. **f** Schematic of the *CFTR* locus and Cas9 guide target (exon 11) and editing efficiency in HAE after Cas9 RNP delivery with each of four shuttle peptides. The same four peptide formulations were applied at [40 μM], with [RNP]: 2.5 μM; [gRNA]: 2.0 μM. Indicated shuttle peptide and Cas9 RNP applied for 3 h; 72 h later, cells processed for Surveyor assay. Asterisks denote genome editing. $n = 3$ donors. Data underlying this figure are provided as Source Data file

We observed that optimized S10, S18, and S85 peptides were more than twice as efficient as the CM18-PTD4 in delivering Cas12a RNP to primary NK cells. This finding encouraged further testing in well-differentiated epithelial cells. Following the application of shuttle peptides with GFP protein or Cas RNPs to the apical surface of HAE, rapid and widespread entry was observed in ciliated, non-ciliated, and goblet cells. Remarkably, protein delivery was also confirmed in human tracheal tissue explants. Achieving robust delivery in hard-to-transduce NK cells, a primary immune cell cultured in suspension, and airway cells, a well-differentiated epithelium, suggest that this peptide

technology represents an advance for targeting other difficult-to-transduce human cells.

We were excited to discover that shuttle peptides could transfer recombinant proteins (GFP, luciferase) and Cas RNPs, formed using *Streptococcus pyogenes* Cas9 (*Sp*Cas9) or *Acidaminococcus* species Cas12a (*As*Cas12a), into mouse airways in vivo. This delivery strategy offers promise for gene editing and may allow modification of accessible progenitor cell types of the airway and alveolar regions[55]. Club cells, α6β4[+] cells, and alveolar type II cells are accessible with luminal delivery[41,56,57]. Furthermore, some proximal airway basal progenitor cells may have membrane

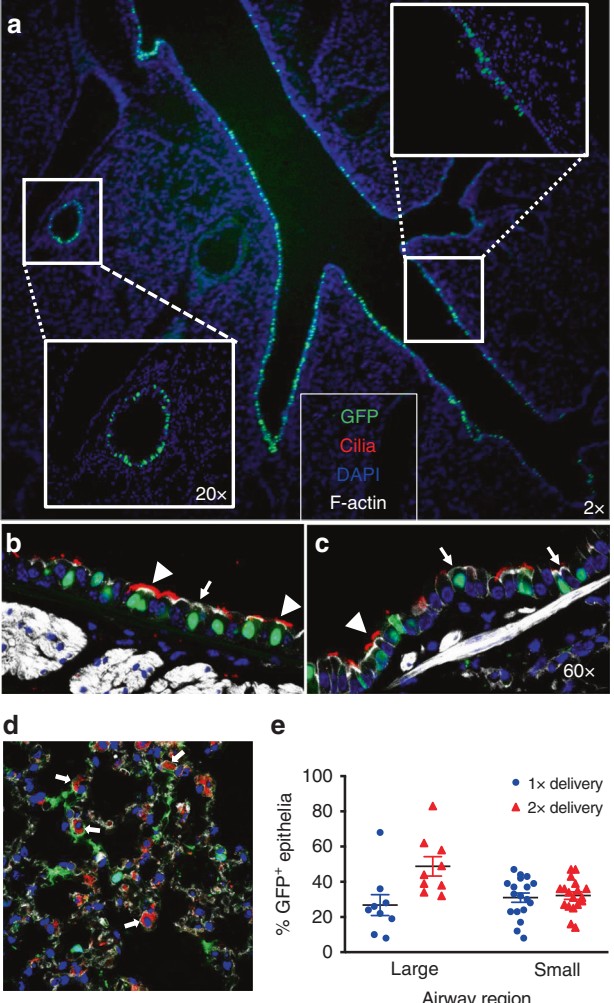

**Fig. 3** GFP-NLS protein delivery to mouse airways using S10 peptide. **a** Fluorescence image of lung tissue section 18 h following two intranasal doses of [S10]: 40 μM; [GFP]: 20 μM in 50 μl; ×2 magnification. Insets show the large and small airways at ×20 magnification. **b**, **c** Localization of GFP in different cell types. GFP co-localized with specific markers of cilia (α-tubulin, red), F-actin (phalloidin stain, gray), and nuclei (DAPI, blue) in large (**b**) and small airway epithelia (**c**). Non-ciliated cells were identified by the absence of α-tubulin staining; ×40 magnification. Arrowheads indicate ciliated cells (α-tubulin); arrows indicate non-ciliated cells. $n = 4$ mice per group. **d** GFP localization in distal lung region. Co-localization of GFP and SP-C (red), a marker of alveolar type II cells, F-actin (phalloidin stain, gray), and nuclei (DAPI, blue); ×40 magnification. White arrows indicate co-localization of GFP and SP-C. **e** Quantitation of GFP$^+$ cells in large and small airways following 1 or 2 deliveries of GFP protein. Results are presented as mean ± SE; $n = 4$ mice per group. Data underlying this figure are provided as Source Data file

extensions that reach the airway lumen[58]. Additionally, repeat administration appears feasible. The shuttle peptides also address some of the inefficiencies and toxicities of non-viral systems. The peptides are quickly degraded[35] and delivery was confined to epithelia of the respiratory tract. The simplicity of the peptide design offers an alternative to small molecules and viral vectors, as the peptides provide both rapid protein delivery and rapid onset of effect. The demonstrated RNP-mediated gene editing may offer therapeutic opportunities for removing aberrant splice sites. One example is CF, where studies examined the relationship between the percentage of cells expressing CFTR and

transepithelial Cl⁻ secretion[59–63]. With relatively good agreement, they suggest that correcting CFTR in 5–15% of cells should restore Cl⁻ secretion to near wild-type levels, editing activity within the range achieved in this study. Additional challenges remain to attain a therapeutic end, including the methods for co-delivery of homologous recombination templates, the identification of alternate CRISPR nucleases or base editors[64,65], and evaluation in larger animal models[38].

The highest editing was observed on epithelial cells using *As*Cas12a, while *Sp*Cas9 RNP achieved limited editing. Interestingly, in vivo *Sp*Cas9 and *As*Cas12a editing were similar. Direct comparisons of nuclease activities are challenging as multiple factors may influence the observed editing efficiency. Indeed, the intrinsic activity of each nuclease may differ, but the delivery efficiencies also greatly influence efficacy. While sharing similar sizes (*Sp*Cas9, 170 kDa and *As*Cas12a, 156 kDa), a major characteristic likely influencing delivery is the net negative charge density contributed by their respective gRNAs. *As*Cas12a, a type V CRISPR protein, uses a simple crRNA (CRISPR RNA) (42 nucleotides (nts)), and *Sp*Cas9, a type II CRISPR protein, requires a crRNA and a tracrRNA (*trans*-activating crRNA) (~100 nts). While the Cas9/crRNA/tracrRNA complex can be delivered using lipids or electroporation in cells responsive to these transfection methods[16,35], the shorter crRNA appears to reduce the Cas12a efficiency using DNA delivery methods. For example, including a DNA carrier in electroporation solutions or increasing the crRNA length both improve Cas12a delivery by some methods[66]. In contrast, we observed that adding the longer cRNA/tracrRNA or Cas9 RNP to a protein cargo reduced shuttle peptide-mediated protein delivery (Supplementary Fig. 6).

In addition to delivery efficiency, the gRNA choice greatly influences Cas12a and Cas9 nuclease activity as the on-target and off-target activities of each individual gRNA can vary widely[67,68]. We note that the indel% observed in ROSA^mT/mG mice based on fluorophore conversion is likely an underestimate as single DNA double-strand breaks at *loxP* sites would not cause fluorophore switching. Finally, as the Cas9 and Cas12a PAMs differ in the design of their gRNA, it is not possible to compare nuclease activities using the same gRNA or the same targeted sequence. Interestingly, to cleave the *loxP* sequence with Cas12a, a CTTC PAM in the *loxP* was used, opening the possibility to target any Cre-lox-based reporter. While effective, this PAM is suboptimal and in vivo activity should improve using a different gRNA with a TTTV PAM[45] or a more permissive engineered Cas12a[69].

A potential limitation of shuttle peptide delivery is cell membrane injury resulting from their endosomolytic sequence region[70]. As critical doses and dosing intervals are established for specific applications, toxicity may be minimized and repeat administration possible. The results of initial toxicity studies (Fig. 5) are encouraging. Despite no evidence of short-term toxicity, future studies should address the immunogenicity of specific therapeutic cargoes[71]. Further investigations will also be required to establish the relationship between the unique physicochemical properties of peptide sequences that provide specific molecular interactions depending upon the nature of protein cargo and target cell type[35,72]. We note that GFP protein and Cas RNP delivery to the airways of mice was heterogeneous. Improvements in the efficiency of shuttle peptide delivery to the respiratory tract might be achieved through screening new engineered peptides, testing formulations, and aerosol delivery.

In summary, this shuttle peptide platform successfully delivered several protein cargoes to human primary NK and airway epithelial cells and achieved clinically relevant genome editing in mouse airways without signs of short-term toxicity. The technology represents a versatile delivery strategy for therapies and to address biological questions. This provides a breakthrough

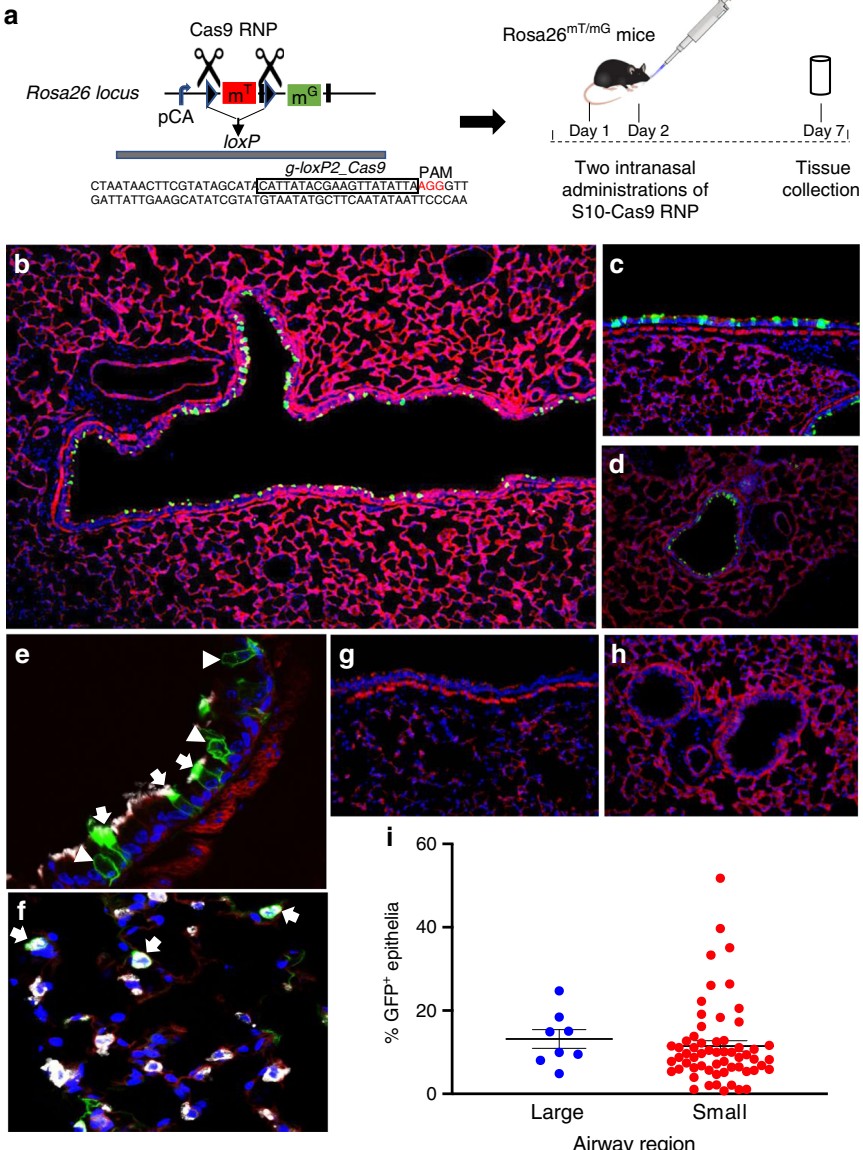

**Fig. 4** S10 peptide delivery of Cas9 RNP shows editing in ROSA^mT/mG locus in vivo. Cas9 RNP directed to *loxP* sites flanking the tdTomato cassette were administered with S10 peptide once daily on 2 consecutive days. Seven days later, conversion of tdTomato to GFP expression was visualized in lung tissue sections. **a** Schematic of gRNA targeting two *loxP* sequences flanking tdTomato gene and experimental protocol for Cas9 RNP delivery to ROSA^mT/mG mice. **b** Fluorescence image of large airway 7 days following two intranasal doses of [S10]: 40 μM; [Cas9]: 1.33 μM; [gRNA]: 2 μM; ×2 magnification. GFP expression denotes edited cells. **c** Editing in a large airway, ×20 magnification. **d** Editing in a small airway; ×20 magnification. **e** Co-localization of GFP and marker of ciliated cells (α-tubulin, white) in large airway. Arrows indicate α-tubulin co-localization with GFP. Arrowheads denote edited (GFP^+) non-ciliated cells negative for α-tubulin; ×40 magnification. **f** Co-localization of GFP^+ and SP-C (white) identifies alveolar type II cells (arrows) in distal lung; ×40 magnification. **g** Representative image of large airway after delivery of Cas9 RNP alone shows no editing, ×20 magnification. **h** Representative image of small airways after delivery of Cas9 RNP alone shows no editing; ×20 magnification. **i** Editing efficiency of Cas9 in large and small airways quantified by the number of GFP^+ cells. Horizontal lines indicate mean ± SE; n = 5 mice/group. Data underlying this figure are provided as Source Data file

approach for hard-to-transduce cell types and may serve as a platform for further design opportunities with other cargoes including therapeutic antibodies and peptides.

## Methods
**Recombinant proteins, synthetic peptides, and gRNAs**. Nuclear-targeted eGFP (GFP-NLS) was expressed in bacteria (*Escherichia coli* BL21DE3) and purified as previously described[35]. *As*Cas12a or AsCpf1 nuclease (catalog #1081069, IDT, Coralville, IA or catalog #A034b-a-0500PMOL, Feldan Therapeutics, Quebec, QC), *Sp*Cas9 nuclease proteins (catalog #1081059, IDT, Coralville, IA or catalog # A037-a-0500PMOL, Feldan Therapeutics, Quebec, QC), and firefly luciferase (catalog # E1708, Promega, Madison, WI) were purchased or provided by Feldan Therapeutics. Peptides were synthetized by GL Biochem (Shanghai, China). Cas9 tracrRNA was purchased from IDT (catalog # 1072532) and specific crRNAs were

synthesized by IDT (Coralville, IA) or Sigma-Aldrich Corporation (St. Louis, MO). Lyophilized peptides were resuspended to 250 μM in PBS and kept frozen. GFP-NLS protein was formulated at 5 mg/ml in PBS and kept frozen. Cas proteins and luciferase were kept frozen in manufacturer buffers. gRNAs were stocked frozen at 100 μM and then diluted at 10 μM for use. Prior to the experiment, Shuttle peptides, GFP-NLS, luciferase, gRNA, and Cas proteins were thawed and kept on ice.

**Cell culture**. Primary airway epithelia isolated from human or porcine donor trachea or bronchi were grown at the air–liquid interface on collagen-coated Costar Transwell polycarbonate filters (#CLS3413, 0.3 μm² surface area)[73]. Cultures were maintained in media supplemented with Ultro-ser G (USG) and the following antibiotics: penicillin (50 U/ml) and streptomycin (50 μg/ml). The cultured cells were maintained at 37 °C in 5% CO₂. All epithelial cells were well differentiated (>4 weeks old; resistance >1000 Ω × cm²). NK cells were kindly provided by Green

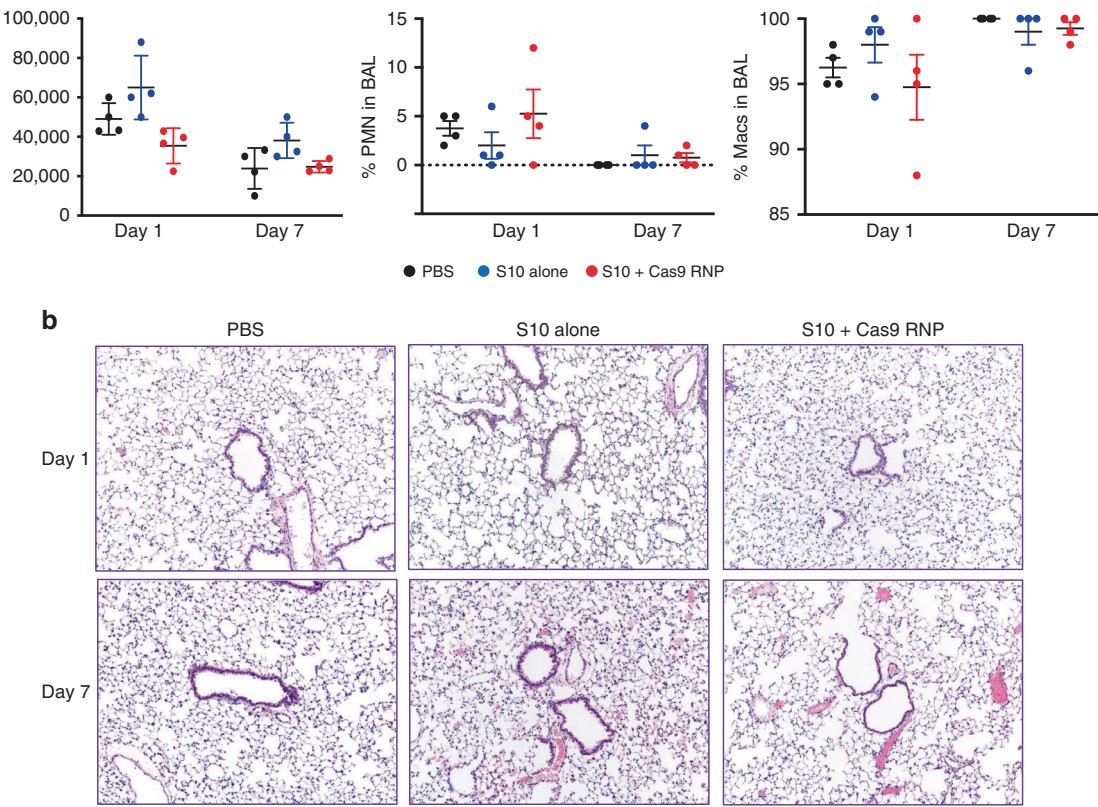

**Fig. 5** Pulmonary toxicity profile following in vivo delivery using S10 peptide. C57BL/6 mice received PBS alone, S10 alone, or [S10]: 40 μM and Cas9 RNP (2.5 μM Cas9/2.0 μM gRNA) by intranasal instillation. **a** In vivo delivery of PBS, S10 alone (40 μM), or S10 (40 μM) and Cas9 RNP (2.5 μM Cas9/2.0 μM gRNA) to C57BL/6 mice by intranasal instillation. Left panel: BAL total cell counts 1 and 7 days post delivery. Middle panel: BAL neutrophil (PMN) percentage 1 and 7 days post delivery. Right panel: BAL macrophage percentage 1 and 7 days post delivery. Each point represents data from one animal. Results represent mean ± SE. **b** Representative images of lung tissue histopathology at 1 and 7 days post delivery. n = 3 mice/group; ×10 magnification. Data underlying this figure are provided as Source Data file

Cross LabCell (South Korea) and cultivated in NK MACS media containing 1% human serum.

**Shuttle peptide-protein formulation**. Shuttle peptides were diluted to obtain the desired final concentration and a volume of PBS was added to achieve the remaining volume minus the volume of the cargo. A volume of protein cargo (GFP-NLS or luciferase) was added to reach the desired concentration. The solution was mixed and added to cells or delivered intranasally to mice.

**Shuttle peptide-Cas RNP formulation**. The RNP was prepared by combining the respective recombinant nuclease protein and the gRNA in PBS and incubating at room temperature for 15–20 min. For SpCas9 delivery, the two-part CRISPR RNA (crRNA and tracrRNA) was first combined in PBS by denaturation at 95 °C for 10 min and renaturation at room temperature to form the RNA duplex (10 μM concentration). Cas RNP complexes were formulated at 2× concentration to a final volume of 50 μl. Shuttle peptides solution (250 μM) was formulated in PBS at 2× concentration in a volume of 50 μl and then mixed with 50 μl Cas RNP mixture. The solution was mixed and added to cells or delivered intranasally to mice.

**Cas12a RNP delivery to primary NK cells**. NK cells were centrifuged for 2 min at 400 × g, washed with warmed PBS, and resuspended at a concentration of 5 × 10[6] cells/ml. Five hundred thousand NK cells were distributed in 1.5 ml tubes and centrifuged for 2 min at 400 × g. Supernatant was removed and cells were resuspended with 50 μl mixture containing the peptide (7.5 μM) and the Cas RNP (Cas12a at 1.33 μM and crRNA at 2.0 μM) for 90 s. Cells were washed and transferred in a 24-well plate with 500 μl of NK MACS media supplemented with 1% human serum and incubated for 48 h.

**GFP-NLS delivery to tracheal tissue explant organ cultures**. Tracheal tissues deemed unsuitable for lung transplantation were used. The excised tissues were bisected longitudinally to expose the luminal surface and then cut into 5 × 5 mm² pieces. Tissue pieces were then placed, lumen side up, onto Costar 48-well, polystyrene culture plates. The explants were immersed in 250–350 μl of airway cell culture media with USG and antibiotics: penicillin (50 U/ml) and streptomycin (50 μg/μl). To transduce the tissues, a volume of 150 μl shuttle peptide (20 μM) formulated with nuclear-targeted GFP protein (20 μM) was applied to the apical surface of an explant for 1 h at 37 °C, 5% CO₂. The formulation was then removed by rinsing three times with PBS. The explant was incubated overnight. At ~24 h post delivery, explants were fixed in paraformaldehyde, sucrose protected, and embedded in OCT.

**Cas RNP delivery to airway epithelia**. The final concentration of nuclease and RNA in the RNP preparation was 2.5 and 2 μM (SpCas9) and 1.33 μM and 2 μM (AsCas12a), respectively. The RNP was combined with the desired concentration of shuttle peptide and incubated for ≤5 min before the addition to cells. A volume of 50 μl of the mixture was left on cell surface for times indicated in text and figure legends. The gRNAs used are listed in Supplementary Tables 1 and 2.

**GFP-NLS delivery to human and porcine airway epithelia**. The apical surface of well-differentiated airway epithelia grown at air–liquid interface was washed twice with PBS. A volume of 50 μl shuttle peptide (20 μM) formulated with nuclear-targeted GFP protein (20 μM) was then applied to the apical surface and incubated for the indicated time. After incubation, the materials were removed and the apical surface was washed thrice with PBS.

**Live cell imaging and image analysis**. Live cell GFP fluorescence was visualized on an inverted tissue microscope Olympus IX70 (Olympus Scientific Solutions Americas Corp, Waltham, MA) and the mean fluorescence intensity was obtained using a SpectraMax i3x fluorescence reader (Molecular Devices, San Jose, CA).

**In vivo bioluminescent imaging**. Luciferase expression was monitored noninvasively using bioluminescence imaging[74]. One hour following delivery of a solution containing luciferase protein combined with the CM18-PTD4 peptide in a 50 μl volume intranasally, mice received 100 μl/10 g of body weight (15 mg/ml in PBS, delivered intraperitoneally) of D-luciferin (Xenogen, Alameda, CA) and were anesthetized by isoflurane inhalation. Approximately 5 min after luciferin injection,

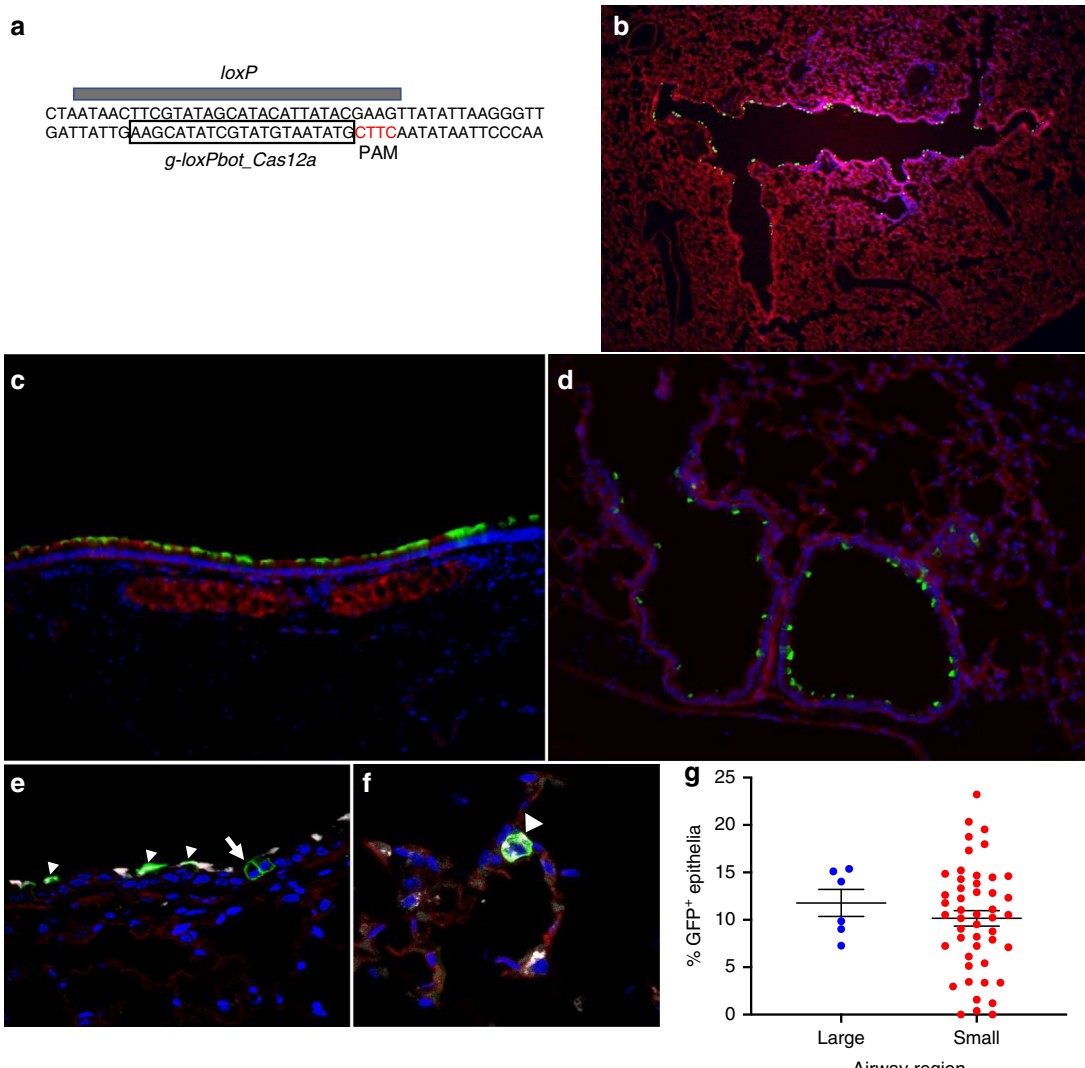

**Fig. 6** S10 peptide delivery of Cas12a RNP shows editing in ROSA^mT/mG locus in vivo. **a** Schematic of Cas12a gRNA targeting both *loxP* sequences flanking tdTomato gene in ROSA^mT/mG mice. S10 peptide and Cas12a RNP (2.5 μM Cas12a; 2.0 μM gRNA) directed to suboptimal PAM target in *loxP* sites flanking the tdTomato cassette administered twice a day for 2 days, via intranasal instillation. Seven days later, conversion of tdTomato to GFP expression was visualized in lung tissue sections. **b** Cas12a mediated deletion of tdTomato in ROSA^mT/mG in large airway epithelia in vivo; ×2 magnification. **c** Editing in a large airway; ×20 magnification. **d** Editing in a small airway; ×20 magnification. **e** Co-localization of GFP and marker of ciliated cells (α-tubulin, white) in large airway. Arrowheads indicate GFP and α-tubulin co-localization. Arrow denotes edited (GFP+) non-ciliated cell negative for α-tubulin; ×40 magnification. **f** Distal lung region. Co-localization of GFP+ and SP-C (white) identifies alveolar type II cells (arrowhead); ×40 magnification. **g** Editing efficiency of Cas12a RNP in large and small airways quantified by the number of GFP+ cells. Horizontal lines indicate mean ± SE; $n = 3$ mice/group. Data underlying this figure are provided as Source Data file

the animals were imaged using a Xenogen IVIS200 CCD camera. Imaging data were analyzed and signal intensity was quantified using the Xenogen Living Image software. Expression levels were normalized by subtracting those from naive (untreated) mice.

**Surveyor nuclease assay**. Three days post transduction, genomic DNA was isolated using QuickExtract (catalog #QE09050, Epicenter) according to the manufacturer's instructions. PCR amplification of the genomic DNA was performed using 700 ng genomic DNA and specifically designed forward and reverse primers (Supplementary Table 3) and KAPA DNA polymerase (KAPA Biosystems). The PCR product was purified and 315 ng of the purified DNA was denatured and reannealed in the presence of NEB restriction enzyme buffer 2 in a thermocycler. The reannealed DNA was mixed with enhancer and Surveyor nuclease (IDT) according to the manufacturer's instructions and incubated at 42 °C for 50 min before separation on 4–20% Tris/Borate/EDTA gels. The DNA bands were visualized using SYBR gold stain (Thermo Scientific).

**T7E1 assay**. The T7 endonuclease I (T7E1) was also used to detect on-target CRISPR-Cas genome-editing events in cultured cells. Briefly, genomic DNA from target cells was amplified by PCR. The PCR products were then denatured and reannealed to allow heteroduplex formation between wild-type DNA and CRISPR-Cas-mutated DNA. T7E1, which recognizes and cleaves mismatched DNA, was used to digest the heteroduplexes. The resulting cleaved and full-length PCR products were visualized by gel electrophoresis. The T7E1 assay was performed using T7E1 (NEB, catalog #M0302S). After the delivery of the CRISPR-Cas complex, cells were lysed in 100 μl of Phusion™ High-Fidelity DNA polymerase (NEB, catalog #M0530S) laboratory with additives. The cells were incubated for 15–30 min at 56 °C, followed by deactivation for 5 min at 96 °C. The plate was briefly centrifuged to collect the liquid at bottom of the wells. Fifty microliters of PCR samples was analyzed for each condition. The PCR samples were heated to 95 °C for 10 min and slowly (>15 min) cooled to room temperature. The PCR product (5 μl) was then separated on a 2% agarose gel to confirm amplification. Fifteen microliters of each reaction was incubated with T7E1 nuclease for 25 min at 37 °C. Immediately, the entire reaction volume was run with the appropriate gel loading buffer on a 2% agarose gel.

**DNA signal intensities quantification**. Bio-Rad ImageLab™ software (Version 5.2.1) was used to quantify the relative signal intensities of each of the bands directly on gels. Quantification of indels was performed using the equation 100 ×

$(1 - (1 - (b + c)/(a + b + c))^{(1/2)}$, where $a$ is the intensity of undigested PCR product and $b$ and $c$ are the intensities of the cleaved products[75]. The sum of the three bands of interest (one gene target and two cleavage products) in a given lane corresponds to 100% of the signal. No cleavage product bands were found in the negative controls.

**Cloning and sequencing to determine editing efficiency**. The PCR products amplified for Surveyor assay were also cloned into the pSMART cloning vector (Lucigen). Purified DNA was phosphorylated with T4 PNK at 37 °C for 30 min before ligation. DNA ligation was performed according to the manufacturer's protocol. The ligation mixture was transformed into DH5α cells, and then plated on LB ampicillin plates. At least 50–100 colonies were grown for plasmid preps and for Sanger sequencing (Functional Biosciences, Madison, WI).

**Immunohistochemistry**. Epithelial cells, tracheal explants, and lung tissues were prepared for histological examination and for immunohistochemistry for cellular markers after fixation in 4% paraformaldehyde, sucrose protection, and cryoembedding in OCT. In some cases, cells and tissues were embedded in paraffin. For routine histology, tissue sections (~8 μm each) were stained with hematoxylin and eosin. A detailed description of methods for localizing protein expression in the large and small airways is provided below.

To co-localize GFP-NLS protein or membrane-associated GFP expression (ROSA$^{mT/mG}$ mice) with markers of specific cell types, sections were first incubated with primary antibodies to acetylated α-tubulin (1:200 dilution, catalog #D2063, Cell Signaling), Muc5AC (1:200 dilution, catalog #MA5-12178, Invitrogen), or SP-C (1:25 dilution, catalog #PA5-71680, Thermo Fisher), and by incubation with an Alexa Fluor 647 phalloidin (1:50 dilution, catalog #A22287, Invitrogen), and mounted with Vectashield with 4′,6-diamidino-2-phenylindole (DAPI) (Vector Labs). Sections were imaged by confocal microscopy using a Leica CTR6500 multiphoton confocal microscope (Leica, Buffalo Grove, IL). Three to five mice (male and female) were analyzed per time point and per experimental condition.

**Flow cytometry analysis**. Cells were lifted from the Transwell support by applying Accumax (catalog #AM105, Innovative Cell Technologies) to the cell surface for 1 h. The cells were pelleted, resuspended in PBS (3% fetal calf serum), and strained through a 70 μm nylon mesh filter. GFP$^+$ cells were enumerated using a Becton Dickinson FACScan, FACS caliber, LSRII (BD Biosciences, Sparks, MD).

**Mice and in vivo delivery**. Shuttle peptide and protein cargo were formulated and delivered to 8–10-week-old, male and female BALB/c or C57BL/6 mice by intranasal delivery. To transduce the epithelia airway, a volume of 50 μl shuttle peptide (40 μM) formulated with nuclear-targeted GFP protein (20 μM) was used. For genome editing, a volume of 50 μl shuttle peptide (40 μM) formulated with nuclease and RNA in the RNP preparation of 2.5 and 2 μM (SpCas9) or 1.33 μM and 2 μM (AsCas12a) were used. Mice were anesthetized with ketamine/xylazine intramuscularly (87.5 mg/kg ketamine/12.5 mg/kg xylazine) and the indicated amphiphilic peptide and protein cargoes introduced intranasally. In some studies, ROSA$^{mT/mG}$ mice (Jackson Labs, Stock #007576) were used[42].

**Determination of GFP$^+$ epithelial cells in vitro**. Epithelial cell sheets were fixed with 4% paraformaldehyde and washed in PBS. The Costar filter inserts were cut from the plastic support, placed onto a microscope slide, and mounted with Vectashield Mounting Media with DAPI. Using a Leica CTR6500 confocal microscope, images of six random fields from each filter were captured at ×20 magnification. Manual counts were performed by enumerating the blue nuclei and GFP$^+$ cells. All epithelial cells within these six fields were counted. For determining the percentage of positive cells, the numerator was the number of GFP$^+$ cells and the denominator was the total number of cells in the six ×20 magnification fields we counted for each epithelial sheet.

**Determination of positive epithelial cells in mouse airways**. Lung tissues were fixed with 4% paraformaldehyde by perfusion and cryopreserved. The tissue was placed in a cryomold en bloc, ventral side down to ensure a uniform and consistent sectioning of all lobes. Tissues were sectioned at 8 μm thickness. We collected sections for study at a depth of sectioning when all lung lobes were visible in the block. These sections provided areas that contained small airways. Next, sections were collected ~150 μm deeper into the block. This region consistently contained large airways with monopodial branching. Consecutive sections were collected for cell counting and co-localization using immunocytochemistry.

The following criteria were used to quantify GFP$^+$ airway cells: because mouse lungs exhibit monopodial airway branching and these airways lack cartilage or glands, they are anatomically defined as bronchioles. Accordingly, we anatomically segregated bronchiolar airways into two groups based on morphologically distinct sites/sizes. The right or left main bronchus was categorized as large airways. Small bronchiole airways were defined as being a terminal airway of similar size. We measured lumen area using ImageJ by tracing along lumen circumference and measuring lumen length and width. All small airways in this study had lumen areas <50,000 μm$^2$ with length-to-width ratios <1.7. All epithelia of the small

airways in one lobe were counted in each area. We counted the total number of cells in the epithelial layer of each airway using the DAPI nuclear stain. We then counted the total number of GFP$^+$ cells in the same airways. The total number of epithelial cells and GFP$^+$ cells in the large airways were counted using the same methods as for small airways.

**Bronchoalveolar lavage**. At 1 or 7 days post instillation of PBS, S10 peptide alone, or S10 and Cas9 RNPs, mice were euthanized by inhalation of CO$_2$, followed by cervical dislocation. The chest was opened and lungs lavaged in situ via PE-90 tubing inserted into the exposed trachea. A pressure of 25 cm H$_2$O was used to lavage the lungs with a total of 4 ml sterile saline. Collected washes were immediately pooled and placed on ice. BAL cell counts were performed using a hemocytometer. Differential cell counts were performed on cytospins following cell staining with a Diff-Quik Stain Set (Siemens Healthcare Diagnostics, Newark, DE).

**Histopathology**. For quantification of cellular infiltrates into lung tissue, a pathologist performing quantification was masked to the group assignment[76]. High-resolution digital images (×100 or ×200 fields) of lung tissues were collected and the number of respective immune cells were enumerated per image field and standardized per image area (mm$^2$). Statistical analysis was performed using Prism (GraphPad Software, La Jolla, CA, USA), and the Mann–Whitney test with statistical significance placed at $P < 0.05$. Tissues were evaluated by an ACVP boarded pathologist and scored using a post examination masking method[76]. Four mice (male and female) were analyzed per time point and per experimental condition.

**Extraction of total RNA and quantitative reverse transcription-PCR**. Lung tissue was homogenized in Trizol and total RNA isolated using the Direct-zol RNA MiniPrep kit (Zymo Research, Irvine, CA). A DNase treatment step was included. Two hundred nanograms of total RNA was used as a template for first-strand complementary DNA (cDNA) synthesis. The resulting cDNA was diluted 4× and subjected to amplification of selected cytokine and chemokine genes by real-time quantitative PCR using Power SYBR Green PCR Master Mix (Applied Biosystems). The primer sequences are listed in Supplementary Table 4. Average values from duplicates of each gene were used to calculate the relative abundance of transcripts normalized to β2-microglobulin and presented as $2^{-\Delta CT} \times 10$ relative to normalizer level. The primers used were previously reported[77].

**LDH assay**. Primary airway epithelial cell cultures from three non-CF human donors were transduced with the S10 peptide alone or co-incubated with GFP-NLS protein in a 50 μl volume for 15 min. The apical surface was washed, and the basolateral media collected at indicated time points post transfection. LDH cytotoxicity assay kit (Cayman chemical) was used to measure the LDH levels in the apical surface washes and basolateral media. Percentage toxicity and viability were computed based on LDH levels. Data were normalized to cells treated with Dulbecco's modified Eagle's medium (DMEM) alone (negative control) or cells treated with DMEM containing 1% Triton X-100 (positive control) on the apical and basolateral surface for 30 min to 10 days at 37 °C.

**Statistical analysis**. Analysis of variance (ANOVA) with Tukey's multiple comparison test or two-way ANOVA with Dunnett's multiple comparison test were used to analyze differences in mean values between groups. Results are expressed as mean ± SE. $P$ values ≤ 0.05 were considered significant.

**Statement on ethics**. Primary airway epithelia and in vivo studies were approved by the Institutional Review Board at the University of Iowa. All animal procedures were approved by the University of Iowa's Institutional Animal Care and Use Committee (IACUC) and were in accordance with National Institutes of Health guidelines. NK cell study was approved the National Research Council Research Ethics Board of Canada.

## Data availability
All relevant data supporting the key findings of this study are available within the article and its Supplementary Information files or from the corresponding authors upon reasonable request. The source data underlying Figs. 1b, d, 2b–f, 3e, 4g, 5a, 6g and Supplementary Figs. 1–4, 5a and 6 are provided as a Source Data file.

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

## Acknowledgements

We thank Jennifer Bartlett, Ashley Cooney, Mario Harvey, Miguel Ortiz, and Patrick Sinn for critically reviewing the manuscript. This work was supported by the National Institutes of Health: 1UG3 HL-147366-01, P01 HL-51670, P01 HL-091842, the Center for Gene Therapy of Cystic Fibrosis: P30 DK-54759, the Cystic Fibrosis Foundation (MCCRAY15XX0). We acknowledge the support of the In Vitro Models and Cell Culture Core, Cell Morphology Core, and the Comparative Pathology Core. P.B.M. is supported by the Roy J. Carver Charitable Trust. Support was also received from the Canadian International Innovation Program (CIIP) of the Global Affairs Canada and delivered collaboratively with the National Research Council as represented by its Industrial Assistance Program (IRAP-898129).

## Author contributions

S. Krishnamurthy, P.B.M. and D.G. conceived and designed the experiments. S. Krishnamurthy, G.S., S. Kandimalla, C.W.-L., D.K.M., V.T., S.H., J.-P.L.-S., X.B. and A.-M.D. carried out experiments and analyzed data. S. Krishnamurthy, P.B.M., T.D., X.B. and D.G. wrote the manuscript.

## Competing interests

P.B.M. is a founder of and holds equity in Talee Bio. D.G. holds equity in Feldan Therapeutics. D.G., T.D., and J.-P.L.-S. hold employee stock options from Feldan Therapeutics. V.T., S.H., A.-M.D., T.D., J.-P.L.-S., X.B., and D.G. are employees of Feldan Therapeutics. D.G., T.D., and J.-P.L.-S. are inventors of US Patents No. 9,738,687 and 9,982,267 related to this technology and assigned to Feldan Bio Inc. The other authors declare no competing interests.
