## [Peer Review File · Nature Communications]

Reviewers' Comments:

Reviewer #1:

Remarks to the Author:

This manuscript by Krishnamurthy et al. describes progress in non-viral delivery of genome editing enzymes to airway epithelial cells in vivo, following initial characterization in vitro. Building off of a previous report from Feldan Therapeutics that established the CM18-PTD4 shuttle peptide, the present work details the ability of shuttle variant sequences to perform transduction of macromolecular cargo including enzymes such as Cas9 and Cas12a. The results are encouraging and indeed demonstrate that shuttle-driven genome editing can induce clinically-relevant levels of genome editing in airway epithelia in mice without causing marked short-term toxicity. This paper is well written, the data are clearly presented, and the conclusions are in accordance with the results. This work represents a substantial advance for the field of non-viral macromolecular delivery (especially that of genome editing enzymes), which has great clinical promise.

The introduction would benefit from a statement regarding the motivating advantages of an RNP-based strategy, e.g. decreased risks (vs. virus, as noted in the discussion) and ease of manufacture (vs. viral or nanoparticle strategies). Perhaps the following sentence, which is currently in the Results section, could simply be moved to the introduction: "RNPs comprised of Cas nucleases and guide RNAs (gRNA) are of interest for therapeutic gene editing because of their rapid effect and transient nuclease activity⁴¹⁻⁴⁴."

It is imperative that quantification of indels detected via gel (following T7E1 digestion) be performed properly. Using densitometry to assess relative band intensities does not provide indel % (and thus does not reflect editing efficiency in a linear and informative manner), yet this is what seems to have been done according to the materials/methods section. Furthermore, the precursor report (Del'Guidice et al. 2018) seems to have committed the same error. Instead, it is appropriate to use the equation $100 \times (1 - (1 - (b+c) / (a+b+c))^{1/2})$, where a is the intensity of undigested PCR product, and b and c are the intensities of the cleaved products (see Guschin et al. (2010) in Engineered Zinc Finger Proteins. It seems that this might only apply to the data in Figure 1b, but it is a crucial distinction.

Throughout, it would be helpful for the figures to append "RNP" with the identity (Cas9 or Cas12a) so it is straightforward to determine which enzyme is being used in each experiment. For example, the data in Figure 2 results from the use of two different enzymes, but this is too easy to overlook. In Figure 5, referring to the peptide as simply "shuttle" is counter-productive and confusing, especially since other figures use the term "peptide". I propose the text "shuttle" in the figure be replaced with "CM18-PTD4 peptide". An analogous improvement could be made to Supplemental Figure 4 (specify the peptide identity in the figure itself). Generally, the reader should be able to know which cargo and which peptide is being specified. In this sense Figure 4 (Cas9 delivered via S10) is an improvement over Figure 6 (Cas12a can be inferred if the reader really tries; no mention of the peptide).

A small box outlining the "legend" section of Figure 3b would improve clarity. The text describing Figures 3e / 4g / 6g would be improved by describing how many sections are represented from each animal. The relationship between the data points presented, the number of animals per group, and the 5 sections (for large airways, as described in the supplement) is not clear. Even a general statement along the lines of "2–5 sections from small airways were quantified per animal" would be helpful.

Regarding the small airway data in Figure 6g, the efficiency is reported to be "10±1% (mean ± SE)". Please double-check that the standard error is merely 1%; with data that spread somewhat evenly across values from 0% to 20%, one might expect a larger standard error value.

In Supplemental Figure 3, a "PBS control" appears but the details of this condition do not seem to

be described in the text, methods, or the figure legend. It mostly draws the attention because it induces more toxicity than many of the experimental conditions. Furthermore, the entire "methods" section of the LDH assay could be improved. Presumably 50 μ L was administered in all conditions, and the 1% Triton-X100 was in a solution of water (not PBS?), but it would be helpful to state these things explicitly.

The details of the formulations for in vivo delivery experiments are currently poorly or indirectly described; these should be stated explicitly. Order of mixing, stock concentrations, and other factors can be important, so these details should be made available. A lot of detail that might apply throughout currently appears under the heading "Cas12a RNP delivery in primary NK cells" - if those details indeed apply broadly (e.g. to in vivo experiments), they should appear under a more general heading.

In the discussion section, consider rephrasing to "engineered shuttle peptides confer effective and non-toxic transfer of protein or Cas RNP into airway epithelia". Consider rephrasing to "With relatively good agreement, these studies suggest".

Reviewer #2:

Remarks to the Author:

In this study, the authors used engineered amphiphilic peptide to deliver recombinant proteins to the airway epithelial cells. They showed that they were able to use this method to deliver GFP to human epithelial cells in vitro. They also showed that they could use this method to deliver CRISPR-Cas/RNP complex to mouse airway epithelial cells in vivo. My comments are listed below:

1. The authors first showed that S10 CPP-ELD peptide can be used effectively to deliver GFP to human airway epithelial cells in vitro. They wrote, "By morphometric analysis, the S10 transduction efficiency ranged from 27 to 35%." It is not clear to be what type of cells are being quantified for transduction (ciliated cells? Goblet cells? Both?). Please clarify.
2. In Figure 2, what is the "non-peptide control"? Was the gRNA/Cas12a used in this control?
3. The authors wrote, "We next evaluated the editing efficiency of Cas12a RNPs at the HPRT1 locus." It is not clear to me what the purpose of this is. Please clarify.
4. In Figure 3e, when the authors quantified the transduction efficiency of GFP by CPP-ELD peptide, it is not clear to me what the calculated transduction efficiency is referring to (ciliated cells? Goblet cells? Both?). Please clarify. It seems that cell-type specific antibodies were not used for this based on the image provided. What is the reason for this?
5. In the experiment shown in Figure 3, the authors delivered the CPP-ELD peptide "once or twice over an 8hr period." Yet, in the experiment shown in Figure 4, the authors delivered the CPP-ELD peptide "one shuttle-RNP dose/day on two consecutive days". Why use two different dosing regimens for these experiments?
6. In Figure 4G, how was the quantification done? What types of cells were included in this quantification? It would be helpful to provide control images of this experiment (where animals only received Cas9 or no treatment).
7. The authors showed that CPP-ELD peptide administration did not lead to any toxicity in mice. However, there is no data on the off-target effects of this delivery method. I would suggest the authors to include some data on off-target effect in the revised manuscript.

Reviewer #1 (Remarks to the Author):

This manuscript by Krishnamurthy et al. describes progress in non-viral delivery of genome editing enzymes to airway epithelial cells in vivo, following initial characterization in vitro. Building off of a previous report from Feldan Therapeutics that established the CM18-PTD4 shuttle peptide, the present work details the ability of shuttle variant sequences to perform transduction of macromolecular cargo including enzymes such as Cas9 and Cas12a. The results are encouraging and indeed demonstrate that shuttle-driven genome editing can induce clinically-relevant levels of genome editing in airway epithelia in mice without causing marked short-term toxicity. This paper is well written, the data are clearly presented, and the conclusions are in accordance with the results. This work represents a substantial advance for the field of non-viral macromolecular delivery (especially that of genome editing enzymes), which has great clinical promise.

We thank the reviewer for these comments.

The introduction would benefit from a statement regarding the motivating advantages of an RNP-based strategy, e.g. decreased risks (vs. virus, as noted in the discussion) and ease of manufacture (vs. viral or nanoparticle strategies). Perhaps the following sentence, which is currently in the Results section, could simply be moved to the introduction: “RNPs comprised of Cas nucleases and guide RNAs (gRNA) are of interest for therapeutic gene editing because of their rapid effect and transient nuclease activity⁴¹⁻⁴⁴.”

Thank you for this suggestion. We moved this sentence from the Results section to the Introduction.

It is imperative that quantification of indels detected via gel (following T7E1 digestion) be performed properly. Using densitometry to assess relative band intensities does not provide indel % (and thus does not reflect editing efficiency in a linear and informative manner), yet this is what seems to have been done according to the materials/methods section. Furthermore, the precursor report (Del’Guidice et al. 2018) seems to have committed the same error. Instead, it is appropriate to use the equation $100 \times (1 - (1 - (b+c) / (a+b+c))^{1/2})$, where a is the intensity of undigested PCR product, and b and c are the intensities of the cleaved products (see Guschin et al. (2010) in Engineered Zinc Finger Proteins. It seems that this might only apply to the data in Figure 1b, but it is a crucial distinction.

We agree with this comment regarding Figure 1b. We re-analyzed the data for Figure 1b as suggested using the equation from Guschin et al. to provide accurate indel %. The Methods section has been modified accordingly and the agarose gels, band intensities, and calculations are provided in the Source Data file in the Figure 1b tab.

Throughout, it would be helpful for the figures to append “RNP” with the identity (Cas9 or Cas12a) so it is straightforward to determine which enzyme is being used in each experiment. For example, the data in Figure 2 results from the use of two different enzymes, but this is too easy to overlook. In Figure 5, referring to the peptide as simply “shuttle” is counter-productive and confusing, especially since other figures use the term “peptide”. I propose the text “shuttle”

in the figure be replaced with “CM18-PTD4 peptide”. An analogous improvement could be made to Supplemental Figure 4 (specify the peptide identity in the figure itself). Generally, the reader should be able to know which cargo and which peptide is being specified. In this sense Figure 4 (Cas9 delivered via S10) is an improvement over Figure 6 (Cas12a can be inferred if the reader really tries; no mention of the peptide).

Thank you for these helpful suggestions. We have modified the text, figure legends, and figures throughout the main manuscript and supplementary information to clarify which peptide and Cas nuclease protein were used.

A small box outlining the “legend” section of Figure 3b would improve clarity.

As suggested, we added a box around the legend for Fig 3.

The text describing Figures 3e / 4g / 6g would be improved by describing how many sections are represented from each animal. The relationship between the data points presented, the number of animals per group, and the 5 sections (for large airways, as described in the supplement) is not clear. Even a general statement along the lines of “2–5 sections from small airways were quantified per animal” would be helpful.

We apologize for any confusion related to these methods. Our original description of the microscopy methods was somewhat difficult to follow. The 5 sections we mentioned in the original methods were used for cell counting and co-localization by immunocytochemistry. For each experimental protocol, we evaluated lung tissue sections from 3-5 animals. We revised the Methods section in the Supplemental Information section under the heading “Quantitative cell and tissue morphometry” to better describe how the morphometry was performed. The Source Data provide the details of the number of cells counted in tissue sections containing large and small airways from each animal.

As we developed the Source Data file, we found that we had omitted some data points in the original Figures 3e, 4g, and 6g. This is corrected in the revised manuscript. The Source Data and figures were corrected accordingly and these edits did not change the experimental outcomes or our analysis and interpretation of the results.

Regarding the small airway data in Figure 6g, the efficiency is reported to be “ $10 \pm 1\%$ (mean \pm SE)”. Please double-check that the standard error is merely 1%; with data that spread somewhat evenly across values from 0% to 20%, one might expect a larger standard error value.

We reviewed these data in detail as we prepared the Source Data files for the figures. The mean \pm SE for the small airway editing efficiency is indeed $10 \pm 1\%$.

In Supplemental Figure 3, a “PBS control” appears but the details of this condition do not seem to be described in the text, methods, or the figure legend. It mostly draws the attention because it induces more toxicity than many of the experimental conditions. Furthermore, the entire “methods” section of the LDH assay could be improved. Presumably 50 μ L was administered in a solution of water (not PBS), but it would be helpful to state these things explicitly.

We apologize for any confusion related to the description of this experiment. The negative control for this experiment was serum free DMEM media, not PBS, and this is corrected in the revised Supplemental Figure 3 and Methods. The positive control, Triton-X100, was suspended in serum free DMEM media. There were no significant differences in LDH release between any of the experimental conditions except for the positive control. The reviewer is correct that materials were applied to the epithelial cells in a 50 μ l volume for all conditions. We revised Supplemental Figure 3, the figure legend, and the Methods section to improve the clarity regarding the technical aspects of these experiments.

The details of the formulations for in vivo delivery experiments are currently poorly or indirectly described; these should be stated explicitly. Order of mixing, stock concentrations, and other factors can be important, so these details should be made available. A lot of detail that might apply throughout currently appears under the heading “Cas12a RNP delivery in primary NK cells” - if those details indeed apply broadly (e.g. to in vivo experiments), they should appear under a more general heading.

Thank you for this helpful suggestion. We have revised the section in the Methods titled “Shuttle peptide-protein formulations” to address this question. For each use, we now describe in more detail how the peptide and protein cargoes were prepared for application to cells or delivery to animals.

In the discussion section, consider rephrasing to “engineered shuttle peptides confer effective and non-toxic transfer of protein or Cas RNP into airway epithelia”. Consider rephrasing to “With relatively good agreement, these studies suggest”.

Thank you for this suggestion. We have changed this sentence to state: “With relatively good agreement, these studies suggest that engineered peptides confer effective and non-toxic protein and Cas9 or Cas12a RNP transfer into airway epithelia *in vitro* and *in vivo*.”

Reviewer #2 (Remarks to the Author):

In this study, the authors used engineered amphiphilic peptide to deliver recombinant proteins to the airway epithelial cells. They showed that they were able to use this method to deliver GFP to human epithelial cells in vitro. They also showed that they could use this method to deliver CRISPR-Cas/RNP complex to mouse airway epithelial cells in vivo. My comments are listed below:

1. The authors first showed that S10 CPP-ELD peptide can be used effectively to deliver GFP to human airway epithelial cells in vitro. They wrote, “By morphometric analysis, the S10 transduction efficiency ranged from 27 to 35%.” It is not clear to be what type of cells are being quantified for transduction (ciliated cells? Goblet cells? Both?). Please clarify.

In our analysis, the numerator was the number of GFP positive cells and the denominator was the total number of cells in the six 20X magnification fields we counted for each sheet of epithelial

cells. Within these fields, we included all epithelial cells present. We did not further enumerate the % of ciliated cells that were positive, the % of secretory cells that were positive, etc.

We apologize for any ambiguity in this description of the results. We revised this sentence as follows: “By morphometric analysis, the S10 transduction efficiency ranged from 27 to 35% for all cell types of the surface epithelium (32 ± 2.0 , mean \pm SE, $n=3$).”

2. In Figure 2, what is the “non-peptide control”? Was the gRNA/Cas12a used in this control?

The no peptide control consisted of the Cas12a RNP (Cas12a protein combined with gRNA) without a delivery peptide. This is indicated in the figure legend.

3. The authors wrote, “We next evaluated the editing efficiency of Cas12a RNPs at the HPRT1 locus.” It is not clear to me what the purpose of this is. Please clarify.

The purpose of this experiment was to evaluate editing efficacy at another target locus with this RNP delivery strategy. In the revised manuscript, we now state, “To investigate the editing efficiency of Cas12a RNPs for another target, we selected the *HPRT1* locus (Fig. 2e).”

4. In Figure 3e, when the authors quantified the transduction efficiency of GFP by CPP-ELD peptide, it is not clear to me what the calculated transduction efficiency is referring to (ciliated cells? Goblet cells? Both?). Please clarify. It seems that cell-type specific antibodies were not used for this based on the image provided. What is the reason for this?

We apologize if our method of analysis was not clear. We counted all positive epithelial cell types in at least two tissue sections from each mouse. In the large and small airways, we saw no GFP protein delivery to non-epithelial cell types. As described in the Supplemental Information in the section titled “Quantitative cell and tissue morphometry”, the numerator was the number of GFP positive cells and the denominator was the total number of cells within the airway region counted.

As stated in the Supplemental Methods section titled “Immunohistochemistry”, we used the following antibodies to co-localize GFP in individual cell types: acetylated alpha-tubulin for cilia (1:200 dilution, catalog #D2063, Cell Signaling), Muc5AC for goblet cells (1:200 dilution, catalog #MA5-12178, Invitrogen), and surfactant protein C for alveolar type II cells (SP-C, 1:25 dilution, catalog #PA5-71680, Thermo-Fisher). We did not use an antibody to a club cell protein.

Note that the two main surface epithelium cell types in the large and small airways of mice are ciliated cells and secretory club cells¹. Goblet cells are very infrequent in healthy lab mice. We deduced that any surface epithelial cell that was GFP positive and not positive for acetylated alpha-tubulin or Muc5AC was a secretory club cell. Our analysis was for the total number of positive cells and we did not further enumerate the % of ciliated cells that were positive, the % of goblet cells that were positive, etc.

5. In the experiment shown in Figure 3, the authors delivered the CPP-ELD peptide “once or twice over an 8hr period.” Yet, in the experiment shown in Figure 4, the authors delivered the CPP-ELD peptide “one shuttle-RNP dose/day on two consecutive days”. Why use two different dosing regimens for these experiments?

We apologize for any confusion regarding these methods. One reason for the difference in the timing of delivery has to do with the different protein cargoes. In the case of GFP-NLS protein (Figure 3), the protein half-life is around 26 hrs. To assess whether there were more GFP positive cells following GFP-NLS protein delivery, we elected to deliver the two doses of protein over a short interval (8 hours) and quantify delivery approximately 18 hr after the second dose. In Figure 4, we delivered Cas9 RNP to ROSA^{mT/mG} mice and quantified GFP positive cells one week after the second dose. Here the half-life of GFP expression was not a critical factor, as the cells continuously produce new protein.

6. In Figure 4G, how was the quantification done? What types of cells were included in this quantification? It would be helpful to provide control images of this experiment (where animals only received Cas9 or no treatment).

The quantification of cells expressing membrane associated GFP in Figure 4g was performed in an identical fashion as the quantification of GFP-NLS protein delivery in Figure 3e (described above). The details of these methods have been revised in the Supplemental Information under the heading “Quantitative cell and tissue morphometry”. Details regarding co-localization of GFP with cell type specific markers are provided in the Supplemental Information under the heading “Immunohistochemistry”. We included the epithelial cells of the small and large airways in our analysis. Of note, we saw no cell types other than epithelial cells showing evidence of GFP expression. When Cas9 RNP was delivered without an amphiphilic peptide, we saw no GFP expression in mouse lung tissue sections. We now include representative photomicrographs of the large and small airways from a ROSA^{mT/mG} mouse that received Cas9 RNP with no peptide (Figure 4g, h).

7. The authors showed that CPP-ELD peptide administration did not lead to any toxicity in mice. However, there is no data on the off-target effects of this delivery method. I would suggest the authors to include some data on off-target effect in the revised manuscript.

The reviewer raises an important point. While it could be interesting to specifically measure the off-target effects of Cas RNP following CPP-ELD peptide delivery, we believe that the selection of Cas nuclease type and the choice of gRNA sequence will largely dictate the frequency of off-target events.

We note that *Streptococcus pyogenes* Cas9 (SpCas9) has already been improved to decrease off-target events by weakening the interactions between Cas9 and the noncomplementary strand². A similar approach was successfully applied to *Staphylococcus aureus* Cas9 (SaCas9)². Despite the difficulties in properly comparing the genome-wide specificities of Cas9 and Cas12a, it has been suggested that the specificity of *Acidaminococcus sp.* Cas12a nucleases (AsCas12a) is similar to high-fidelity SpCas9 variants³. Furthermore, in the same publication, GUIDE-seq analysis and targeted deep sequencing for both AsCas12a and *Lachnospiraceae*

bacterium Cas12a (LbCas12a) showed no detectable off-target events with different crRNAs³. It was further demonstrated in another study using sustained expression of SpCas9 and gRNA in mouse liver from an adenoviral vector that appropriately designed guide RNAs can lead to effective editing with no detectable off-target mutations⁴. Our use of Cas RNP rather than coding mRNA or DNA (plasmids or viruses) offers the advantages of limiting exposure of the genome to editing machinery and decreasing off-target events⁵.

Since the CPP-ELD derived peptides facilitate delivery of Cas RNP rather than coding mRNA or DNA (plasmids or viruses), are not limited to one type of Cas nuclease, and can accommodate different gRNAs, we think that any off-target events detected in such an experiment will not provide new information and that any potential off-target events revealed may be minimized by using high fidelity nucleases and/or appropriately designed gRNAs.

References

1. Parent, R.H. *Comparative Biology of the Normal Lung*, (Elsevier Inc., London, 2015).
2. Slaymaker, I.M., Gao, L., Zetsche, B., Scott, D.A., Yan, W.X. & Zhang, F. Rationally engineered Cas9 nucleases with improved specificity. *Science* **351**, 84-88 (2016).
3. Kleinstiver, B.P., Tsai, S.Q., Prew, M.S., Nguyen, N.T., Welch, M.M., Lopez, J.M., McCaw, Z.R., Aryee, M.J. & Joung, J.K. Genome-wide specificities of CRISPR-Cas Cpf1 nucleases in human cells. *Nat. Biotechnol.* **34**, 869-874 (2016).
4. Akcakaya, P., Bobbin, M.L., Guo, J.A., Malagon-Lopez, J., Clement, K., Garcia, S.P., Fellows, M.D., Porritt, M.J., Firth, M.A., Carreras, A., Baccega, T., Seeliger, F., Bjursell, M., Tsai, S.Q., Nguyen, N.T., Nitsch, R., Mayr, L.M., Pinello, L., Bohlooly, Y.M., Aryee, M.J., Maresca, M. & Joung, J.K. In vivo CRISPR editing with no detectable genome-wide off-target mutations. *Nature* **561**, 416-419 (2018).
5. Liang, X., Potter, J., Kumar, S., Zou, Y., Quintanilla, R., Sridharan, M., Carte, J., Chen, W., Roark, N., Ranganathan, S., Ravinder, N. & Chesnut, J.D. Rapid and highly efficient mammalian cell engineering via Cas9 protein transfection. *J. Biotechnol.* **208**, 44-53 (2015).

Reviewers' Comments:

Reviewer #1:

Remarks to the Author:

The revised manuscript is much improved. I have no substantial concerns or constructive input. I stand by my previous assessment of the value and importance of this work.

Reviewer #2:

Remarks to the Author:

The authors have adequately addressed my comments in the revised manuscript. It is my opinion that this manuscript is acceptable for publication.

Responses to Review Comments

REVIEWERS' COMMENTS:

Reviewer #1 (Remarks to the Author):

The revised manuscript is much improved. I have no substantial concerns or constructive input. I stand by my previous assessment of the value and importance of this work.

Reviewer #2 (Remarks to the Author):

The authors have adequately addressed my comments in the revised manuscript. It is my opinion that this manuscript is acceptable for publication.

We thank the reviewers for their kind comments.